# Dilemmas in Elderly Diabetes and Clinical Practice Involving Traditional Chinese Medicine

**DOI:** 10.3390/ph17070953

**Published:** 2024-07-16

**Authors:** Chongxiang Xue, Ying Chen, Yuntian Bi, Xiaofei Yang, Keyu Chen, Cheng Tang, Xiaolin Tong, Linhua Zhao, Han Wang

**Affiliations:** 1Graduate School, Beijing University of Chinese Medicine, Beijing 100029, China; xue0911@bucm.edu.cn (C.X.); chenying1001@bucm.edu.cn (Y.C.); yangxiaofei2023@bucm.edu.cn (X.Y.); 2Institute of Metabolic Diseases, Guang’anmen Hospital, China Academy of Chinese Medical Sciences, Beijing 100053, China; 18061591518@163.com (K.C.); tongxiaolin@vip.163.com (X.T.); 3Department of Integrative Cardiology, China-Japan Friendship Hospital, Beijing 100029, China; 4School of Traditional Chinese Medicine, Hunan University of Chinese Medicine, Changsha 410208, China; 15698175139@163.com; 5National Key Laboratory of Efficacy and Mechanism on Chinese Medicine for Metabolic Diseases, Beijing University of Chinese Medicine, Beijing 100029, China; tangcheng0719@bucm.edu.cn; 6School of Chinese Materia Medica, Beijing University of Chinese Medicine, Beijing 100029, China

**Keywords:** traditional medicine, natural products, elderly diabetes, management, clinical evidence, dilemmas

## Abstract

Diabetes is a widespread chronic disease that occurs mainly in the elderly population. Due to the difference in pathophysiology between elderly and young patients, the current clinical practice to treat elderly patients with anti-diabetes medications still faces some challenges and dilemmas, such as the urgent need for early diagnosis and prevention, and an imbalance between restricted dietary intake and the risk of undernutrition. Traditional Chinese medicine (TCM) offers various treatment regimens that are actively utilized in the field of diabetes management. Through multiple targets and multiple pathways, TCM formulas, medicinal herbs, and active natural products enhance the efficacy of diabetes prevention and diabetes control measures, simplify complex medication management, and improve common symptoms and common diabetic complications in elderly people. Historically, natural products have played a key role in material composition analysis of TCM and mechanism interpretation to enable drug discovery. However, there have been few conclusions on this topic. This review summarizes the development of TCM for the prevention and management of diabetes in elderly people, existing evidence-based clinical practices, and prospects for future development.

## 1. Introduction

According to the 10th edition of the International Diabetes Federation’s Diabetes Atlas (https://diabetesatlas.org/atlas/tenth-edition/, accessed on 1 May 2024), approximately 536.6 million individuals worldwide were afflicted with diabetes in 2021, constituting 10.5% of the global population. By 2045, the prevalence is projected to reach 12.2%, encompassing 783.2 million individuals [1]. Global expenditure on diabetes-related health has notably increased, increasing from USD 232 billion in 2007 to USD 966 billion in 2021, among adult patients, signifying a 316% increase over a 15-year time span. China, as a populous nation, has a substantial diabetes burden, affecting not only its general populace, but also its elderly population. According to data from China’s seventh national population census, the incidence of elderly individuals with DM (≥60 years) increased significantly to 30%, impacting 260 million people in 2020 [2,3]. An estimated 45% to 47% of older adults are prediabetic [3,4].

The elderly population is at increased risk of diabetes, making it a key demographic for diabetes prevention and management [5,6]. Elderly individuals with diabetes often experience asymptomatic hyperglycemia, worsened metabolic dysregulation, occult microvascular and macrovascular complications, and heightened susceptibility to acute adverse events [7]. Given the diverse pathophysiological factors associated with aging, the management strategies for elderly patients with diabetes differ significantly from those for younger individuals. The existence of conflicting treatment options in relation to elderly patients with diabetes exacerbates the challenges faced in treatment. Furthermore, insufficient attention is currently being devoted to addressing the unique needs of elderly individuals with diabetes [8,9]. It is imperative to prioritize a comprehensive approach that integrates individualized care, effectiveness, and safety.

For millennia, traditional Chinese medicine (TCM) has been utilized in China, demonstrating efficacy in alleviating symptoms and enhancing quality of life, with notable treatment adherence and safety [10,11]. TCM offers a distinctive perspective on the etiology and progression of conditions such as diabetes and aging, including imbalances in Yin and Yang, disruptions in the flow of Qi and Xue, and the dysfunction of ZangFu, which symbolize the internal organs and their respective functions [12]. For instance, Qi-tonifying herbs could promote energy metabolism to control weight and boost immunity against diabetic complications. Moreover, fundamental research has been undertaken to investigate the systematic anti-diabetes and anti-aging mechanisms of TCM formulas, medicinal herbs, and active natural products at cellular and molecular levels [13,14,15]. Clinical trials involving elderly individuals with diabetes and common diabetic complications have demonstrated the benefits of TCM in terms of three-level prevention, exercise guidance, symptom management, weight, nutrition, and complication management [15]. To the best of our knowledge, this study represents the first attempt to compile a comprehensive review of the challenges faced by elderly individuals with diabetes and to summarize the clinical evidence pertaining to TCM and relevant natural products. The primary objective of this investigation is to present a critical analysis of the clinical application of TCM and natural products in the management of diabetes among the elderly population, encompassing the present status, clinical considerations, existing TCM strategies, and future perspectives in this field.

## 2. Lack of Timely Diagnosis and Inadequate Prevention of Diabetes in Elderly People

Increasing age is one of the risk factors for diabetes [6]. For elderly people with diabetes risk factors, the purpose of prevention is to intervene in abnormal glucose tolerance, reduce the rate of diabetes, reduce the incidence of complications, reduce the incidence of adverse reactions and mortality, and improve the quality of life [9]. Given the elevated incidence of diabetes among elderly individuals, recommendations have been made for early prevention, diagnosis, and treatment [5]. Nevertheless, the current strategies for preventing diabetes in elderly people are inadequate, with a prevailing trend toward ineffectiveness and pessimism [5,9]. Due to the presence of atypical symptoms and numerous complications, the rate of diagnosis and treatment for elderly individuals with diabetes is regrettably low. Additionally, elderly patients with diabetes who present with acute and complex symptoms are prone to misdiagnosis and underdiagnosis. For the secondary prevention of diabetes, we have summarized related potential preclinical and clinical interventions in a previous review [16]. The crosstalk network between metabolic disorders, immuno-inflammation, and endothelial dysfunction may provide novel and effective therapeutic targets for DVC prevention, while TCM and natural product interventions, such as rhein, hirudin, and polysaccharides, still deserve further study because they have unique advantages for DVC prevention. The primary prevention of diabetes may be a more pressing emergency that we should effectively address. Thus, we further reviewed publications of meta-analyses and systematic reviews that shed new light on the primary prevention of DM (Table 1 and Appendix A).

Cognitive decline in individuals with diabetes was highlighted, with a reduced ability to live independently, reduced compliance, reduced effects of lifestyle interventions, and offsets in terms of the endeavor toward diabetes education and the prevention of diabetes [26]. The mechanisms include neuroinflammation as a result of chronic inflammation in elderly people with diabetes, which increases the risk of central degenerative diseases and psychiatric disorders [27,28]. A systematic review of 17 studies, involving 1.7 million patients, revealed that people with diabetes had an approximately 2.25-fold greater risk of developing Alzheimer’s disease than the people without diabetes [29]. The treatment of cognitive impairment in elderly patients with diabetes poses a significant challenge due to a lack of early identification of risk factors, such as hypoglycemia [30]. Addressing issues such as poor compliance, public health education, and health promotion is crucial in managing this population. It is imperative for elderly people to engage in and assimilate into society to mitigate and postpone cognitive decline [31]. The elderly population demonstrates a low health education acceptance rate, and further research from government and society is needed to address the societal impact and policy implementation related to elderly diabetes [32,33,34]. It is essential to advocate for the prevention and rehabilitation of DM in elderly people, with the involvement of family members to cater to the diverse needs of elderly individuals. Extensive evidence from experimental data has illustrated that TCM is effective in the treatment of diabetic cognitive decline, with few adverse effects [35,36]. Evidence-based TCM interventions for diabetic cognitive decline are relatively insufficient (Table 2 and Appendix A).

## 3. Nutritional Imbalances and Restricted Dietary Intake in Elderly Patients with Diabetes

Achieving a balanced diet and proper nutrition is a desired outcome, while overnutrition issues such as being overweight, obesity, and inadequate nutrition-induced frailty and sarcopenia significantly impact elderly patients with diabetes [41,42]. Over 50% of elderly individuals with diabetes are diagnosed with obesity or as being overweight, and data from the Beijing Longitudinal Study of Aging II (BLSA-II) indicate a high prevalence (19.32%) and incidence (12.32%) of frailty in this population [43,44]. A review of epidemiological studies in Asian countries, using the AWGS 2014 standard, revealed that the prevalence of sarcopenia ranged from 5.5% to 25.7% [45].

On the one hand, restricted dietary intake helps individuals lose weight, control blood glucose, and even exert anti-aging effects [46,47,48,49]. On the other hand, adequate nutrition is a necessary guarantee to prevent and control adverse consequences related to malnutrition in elderly individuals [50]. However, a diet that is too heavy or restricted may not be appropriate for the elderly population. It is advisable that elderly individuals enhance their nutritional and protein consumption to maintain healthy dietary practices, avoid overly restrictive energy intake, prioritize a well-rounded diet with balanced nutrition, favor carbohydrates with a low glycemic index, ensure adequate intake of high-quality protein, be vigilant of malnutrition in elderly individuals with diabetes, routinely utilize nutritional risk screening tools for assessing nutritional risk, and aim for the prompt detection of risk factors and interventions to enhance patient outcomes [5,51]. To achieve and maintain a healthy weight, it is important to approach weight loss in a methodical manner that includes a balance of fat loss and muscle gain. This necessitates a thorough examination of weight loss, dietary interventions, and protein intake in elderly patients with diabetes. Sarcopenia is intricately linked to the aging process, as elderly individuals experience alterations in organ function and hormone levels that contribute to diminished exercise capacity, skeletal muscle mass, muscle strength, and overall physical function [52]. Older diabetes patients exhibit an extended disease duration, characterized by heightened levels of advanced glycation end products and reactive oxygen species, resulting in a reduction in capillary density, type II muscle fibers, and impaired muscle synthesis within the muscle tissue [53]. Insulin resistance in elderly individuals with diabetes mellitus hastens the onset of sarcopenia [54]. Malnutrition, which leads to decreased muscle protein synthesis, serves as a significant etiological factor and robust indicator for the progression of sarcopenia. Dietary interventions in isolation often fail to supply sufficient nutrients for elderly patients with sarcopenia, necessitating the potential consideration of oral supplementation. Furthermore, the consumption of proteins rich in essential amino acids is typically advised [55]. However, individuals with diabetic nephropathy may face unique considerations in this context. The dietary guidelines for individuals with diabetes and kidney disease suggest a focus on consuming high-quality protein, while limiting the intake of plant-based proteins [56]. It is important to be cautious of potential adverse reactions in the gastrointestinal tract when using anti-diabetes medications, such as glucagon-like peptide-1 receptor agonists, as they may lead to malnutrition, sarcopenia, and frailty [57,58,59]. Part 4 of this study delves into further precautions and adverse effects associated with anti-diabetes medications. For TCM interventions, acupuncture has been proposed as a potential treatment for obesity and diabetes, and a few high-quality clinical meta-analyses and systemic reviews have reported on the use of TCM and natural products for treating obesity, frailty, and sarcopenia in elderly populations [60,61,62,63]. The systemic reviews and meta-analyses on TCM and natural products for the treatment of obesity, frailty, and sarcopenia are shown in Table 3 and Appendix A.

## 4. Sports Health and Safety Are Equally Important for Elderly People with Diabetes

The dilemma involving sports health and safety for elderly people with diabetes is also a problem that needs to be solved. Falls are common adverse events for older adults. According to WHO statistics (https://www.who.int/publications/i/item/9789241563536, accessed on 1 May 2024; https://www.who.int/news-room/fact-sheets/detail/falls, accessed on 1 May 2024), the incidence of falls in elderly individuals aged 65 years and above is 28~35% worldwide, and for those aged 70 years and above, it is as high as 32~42%. Each year, an estimated 684,000 individuals die from falls globally; thus, falls and related injuries are serious obstacles to healthy aging. Older individuals need to engage in consistent and moderate physical activity, select activities that align with their physical capabilities and health status, and incorporate balance and flexibility exercises into their routine. Prioritizing the safety of exercise for elderly individuals is crucial for preventing the worsening of health conditions, resulting from falls and prolonged bed rest. TCM fitness training, which includes Tai Chi, Ba Duan Jin, and Wu Qin Xi, has been demonstrated to be beneficial for promoting physical stability and decreasing the likelihood of falls [93,94]. Thus, we reviewed the systemic reviews and meta-analyses on TCM exercise for the treatment of diabetes and related complications in elderly individuals (Table 4).

## 5. Medication Safety in Blood Glucose Management for Elderly People with Diabetes

Diverse anti-diabetes medications exert blood glucose regulatory effects through different mechanisms. For various comorbid diseases and special conditions in the elderly population, diabetes control regimens become more complex, and medication application becomes tricky. Anti-diabetic therapeutic regimens that avoid the risk of hypoglycemia and underweight people, allow elderly patients with diabetes to adjust individualized medication plans, while ensuring medication safety [115].

TCM also has significant advantages in decreasing blood glucose with medication. In addition to TCM exercise, an increasing number of TCM prescriptions play a good alternative role, some of which even have high-quality clinical evidence (Table 5). Although the anti-diabetic effects of numerous TCM interventions have been confirmed, few studies have focused on related high-quality clinical evidence in elderly people. Zhang and her colleague conducted the only meta-analysis to survey the evidence on TCM and natural products for blood glucose control in elderly patients with diabetes, providing new insights for future clinical practice [40].

## 6. Multiple Complicated Diseases Make Medication Management More Complex for Elderly Patients with Diabetes

Elderly patients often have complex underlying medical conditions. Polymedication is common and difficult to avoid in elderly patients with diabetes, making matters worse [116]. Decreased liver and kidney function in elderly individuals, coupled with the use of multiple medications, increases the risk of drug interactions and adverse reactions [117]. To mitigate these risks, it is imperative for elderly individuals to adhere to safe, rational, and standardized medication practices, following their healthcare provider’s guidance for precise drug use. Dosage adjustments should be tailored to the individual [9,118]. It is imperative for family members to assist in enhancing medication adherence, by ensuring that medications are taken at the prescribed dosage and on schedule [119]. The management of multiple medications can be intricate, particularly for patients with poor compliance and cognitive decline. Elderly patients with diabetes mellitus frequently suffer from comorbidities necessitating multiple therapeutic interventions, underscoring the importance of vigilance and comprehension regarding drug interactions and effects, to mitigate the risk of inappropriate medication use. Due to its multitarget properties, TCM and natural products have demonstrated efficacy in treating a range of concurrent conditions, such as geriatric cardiometabolic and systemic metabolic disorders [120,121].

### 6.1. Elderly Patients with Diabetes Combined with Cardiovascular Disease (CVD)

Diabetes mellitus is a significant comorbid condition associated with CVD and is the primary cause of morbidity and mortality in individuals with T2DM [122,123]. Individuals with diabetes have a 2–3 times greater risk of developing CVD than those without diabetes [124]. Diabetes also serves as a notable risk factor for heart failure (HF), with a 22.3% greater prevalence of HF and a 56% greater prevalence of hospitalization in elderly diabetes patients [125,126]. Furthermore, older individuals with diabetes have a 10-fold greater risk of mortality from HF than those without diabetes [127]. In addition, for every 1% increase in glycosylated hemoglobin, the risk of heart failure increases by 8~36%, and the risk of HF in diabetic patients increases with age, coronary heart disease, and peripheral vascular disease [128]. TCM and natural products have been utilized in a series of beneficial attempts in this regard, and the related evidence is summarized in Table 6.

### 6.2. Elderly Patients with Diabetes Combined with Metabolic Syndrome (MS)

The prevalence of MS differed by 48.91%, according to the International Diabetes Federation criteria, and 46.80%, according to the ATP III criteria [135]. The elevated prevalence of metabolic syndrome among elderly individuals with diabetes mellitus, particularly in older females, may be attributed to alterations in hormone levels post-menopause and diminished metabolic function [136,137]. Tailored treatment for hypertension is recommended for the majority of elderly diabetic patients, with a focus on individualized target levels [138]. In elderly patients with diabetes and hypertension, despite being at increased risk, the blood pressure management goal can be adjusted to 140/90 mmHg [9,139]. Given the significant variability in blood pressure, ambulatory blood pressure monitoring is recommended. The prevention of hypoglycemia and hypotension, along with the associated complications, should be prioritized when managing diabetes and hypertension in the elderly population. Research indicates that lipid-lowering therapy may decrease cardiovascular risk among older individuals. It is imperative to consider the balance between the benefits and risks, potential drug interactions, adverse effects, and individual preferences for lipid-lowering agents [140,141]. This decision should be informed by a thorough evaluation of factors such as life expectancy, frailty, comorbidities, liver and kidney function, and economic considerations.

TCM and natural products can reduce a patient’s weight, lower their blood pressure, and lower their lipids. There is some evidence from basic research and small sample clinical studies to verify the therapeutic effect of TCM and natural products on metabolic syndrome. However, only one single clinical systematic review was identified on TCM and natural products for diabetes in conjunction with metabolic syndrome, specifically examining the effects of *Crocus Sativus* L. on metabolic profiles in patients with diabetes mellitus or metabolic syndrome [142]. The effectiveness and safety of treatments for diabetes and metabolic syndrome are still uncertain due to the limited quality and heterogeneity of the existing studies on the topic.

## 7. TCM and Natural Products Have Advantages in Terms of the Management of Common Symptoms in Elderly Patients with Diabetes

The efficacy of TCM and natural products in managing complex symptoms is a notable feature over that of Western medicine [143,144]. However, the existing evidence supporting the use of TCM and natural products for common symptoms, such as urinary incontinence, dizziness, falls, constipation, and frailty, is lacking, and related research and clinical practices are mostly based on empirical treatment. There is a dearth of systematic reviews and meta-analyses on the application of TCM and natural products for these symptoms in elderly individuals with diabetes. However, we did not search for any registered trials on common symptoms in elderly patients with diabetes. More clinical trials need to be conducted to demonstrate the potential benefits of TCM and natural products in this population. It is evident that TCM offers advantages in managing multiple symptoms, including those that may not be effectively addressed by Western medicine. Throughout the treatment regimen, TCM practitioners tailor comprehensive interventions to the individual patient’s constitution and symptoms, aiming to restore the balance of Yin and Yang, enhance the circulation of Qi and blood, and effectively manage diabetes. Additionally, dietary control, physical activity, and regular monitoring of blood glucose levels are crucial components of diabetes management for individuals receiving TCM and natural product treatments [145,146].

## 8. TCM Offers a Complementary Solution to the Management of Diabetes in Elderly People and Common Diabetic Complications in Elderly People

The existing therapeutic approaches for managing refractory complications in diabetes patients are limited. Li et al. [147] conducted a comprehensive review of the cardiovascular, renal, and retinal outcomes associated with various anti-diabetic interventions in individuals with diabetes mellitus. The interplay of individual patient characteristics frequently plays a crucial role in influencing the adherence and adjustment of chronic kidney disease patients to the ramifications of heightened medication regimens. These factors frequently pose challenges in medical management, resulting in diminished renal function and general debility [148]. Additionally, diabetic panvascular diseases, including diabetic retinopathy (DR), diabetic cardiomyopathy (DC), diabetes-related coronary heart disease (CHD), and diabetic peripheral neuropathy (DPN), present as troublesome complications. Pioneering TCM research has been conducted to address these complications, and we provide a review of the related evidence in Table 7.

## 9. Challenges and Future Perspectives

The favorable acceptance rate of TCM and natural products among the elderly population in China underscores the imperative for the establishment and dissemination of population-specific prevention strategies. Ethnopharmacology-focused research endeavors not only offer a scientific foundation for determining optimal dosages and potential toxicological impacts within local communities, but also hold promise for the development of more efficacious multitarget pharmaceuticals aimed at preventing and treating a range of ailments, including diabetes, in elderly people. Learning from the successful research experience involving Artemisinin, a selection of effective drug targets of TCM and natural products under the guidance of unique TCM theory accelerates the drug discovery process. TCM and natural products also play a role in enhancing the effectiveness of anti-diabetes medications and in the prevention and management of complications, offering a novel perspective. In the process of drug discovery, a crucial mechanism of diabetes provides directions for target selection. For example, the conceptualization of gluco–cardio–renal conditions helps to guide researchers to explore multitarget anti-diabetes medications, while that is exactly what TCM is good at. By integrating various disciplines, including epidemiology, clinical, and experimental monitoring, we conducted a comprehensive investigation and analysis of the clinical dilemmas in elderly diabetes (Figure 1). This study aims to establish a robust scientific foundation for clinical practice. Following a thorough examination of diabetes in the elderly population, specific intervention strategies are implemented, and their effectiveness in enhancing patients’ quality of life and mitigating the occurrence of complications and mortality is assessed.

TCM and natural products have multiple pathways and multitarget metabolic regulatory properties. Highly potent chemical compounds used for elderly diabetes are summarized and their chemical structures are shown in Figure 2. Although numerous achievements have been made regarding TCM and natural products in regulating glucose and lipid metabolism, there still needs to be further in-depth research in this regard and there is still a long way to go for the application and translation of TCM and natural products into the clinical field. We have to admit that the biological characterization of TCM and natural products for elderly diabetes is at a very early stage, most of them are not fully characterized in vitro and in vivo. Also, the shortage of TCM and natural products in regard to various aspects such as low solubility, low bioavailability, and low tissue targeting, need to be overcome. The following suggestions are possible research directions with value based on TCM and natural products for treating disorders in elderly diabetes.

(1)The development of novel elderly diabetes patient-centered TCM and natural product treatments. Search for more effective TCM and natural products to enhance the therapeutic effect;(2)Transition from experience-based to evidence-based approaches in TCM and natural products. Facilitate the establishment of guidelines and promote international collaboration in addressing elderly diabetes patients;(3)Clinical trials: Carry out large-scale clinical trials to encompass a more representative population of cases and controls. Enhance the dependability and precision of research findings, as well as delving deeper into the variations in the effectiveness of anti-diabetic medications;(4)Explore a wider range of study methodologies beyond relying solely on RCTs and case reports. Additional clinical evidence, such as real-world studies, may provide a more accurate representation of the clinical efficacy of TCM;(5)Employ novel evaluation standards tailored to the unique features of TCM. Investigate the correlation between TCM syndromes, treatment protocols, TCM applications for specific-patient populations, and utilize artificial intelligence in TCM practice;(6)Personalized treatment: A comprehensive examination of factors, including drug mechanisms of action, drug metabolism, and genetic polymorphisms. Elucidate the discrepancies in drug efficacy and develop personalized treatment plans based on the patient’s genotype and metabolic characteristics;(7)Multidisciplinary and multilevel research approaches to support the prevention and treatment of diabetes in elderly patients. Involve various medical disciplines, such as cardiovascular, nephrology, neurology, sports medicine, and nutrition, in the development of comprehensive treatment plans;(8)The overseas development of TCM. TCM-dominant diseases need more attention to promote the inheritance and innovative development of TCM under a clinical value-oriented research model. To enlarge the influence and recognition of TCM globally, collaboration with modern medicine and the development of acceptable TCM therapy (especially acupuncture, well-defined oral or topical TCM and natural products, TCM exercise) is needed to help build trust in TCM among the general public and healthcare professionals;(9)Research on the active pharmacodynamic material basis of TCM is needed. Numerous natural products from plants or synthesis methods make the composition of TCM definite and clear. Moreover, there is a need to ascertain the applicability of current standards for assessing the clinical efficacy of modern medicine to TCM and natural products;(10)More comprehensive and high-quality verification experiments are needed to test the crucial effects of the hub genes selected in this study, to draw more precise and credible conclusions.

## 10. Conclusions

In comparison to younger individuals, elderly patients exhibit decreased tolerance to blood glucose exposure, diminished tissue and organ reserves, and elevated susceptibility to medication-related risks. TCM practices in China offer a novel perspective on this matter. Active natural products have played a key role in material composition analysis of TCM and mechanism interpretation to enable drug discovery. This review highlights the significant therapeutic potential of TCM and natural products in managing diverse manifestations of diabetes in the elderly population. And there is a clear conclusion that TCM and natural products could: (1) improve the prevention of diabetes in elderly individuals, (2) maintain nutritional balance and sport health, (3) manage clinical symptoms, (4) control and even reverse diabetes and diabetic complications, and (5) provide a complementary and alternative therapy. As for the selection of the best treatment, a well-designed network meta-analysis may provide the final answer. Also, experimental studies are currently being conducted to investigate the efficacy of TCM and natural products in treating elderly patients with diabetes, with a focus on elucidating the underlying mechanisms involved. Future research efforts should prioritize the integration of high-quality clinical evidence and the exploration of novel pharmacological insights related to natural products for the management of diabetes in the elderly population. It is imperative to develop a comprehensive, integrative, and individualized treatment approach for elderly patients with diabetes, with the utilization of TCM and natural products playing a crucial role in optimizing clinical outcomes and maximizing therapeutic benefits.

## Figures and Tables

**Figure 1 pharmaceuticals-17-00953-f001:**
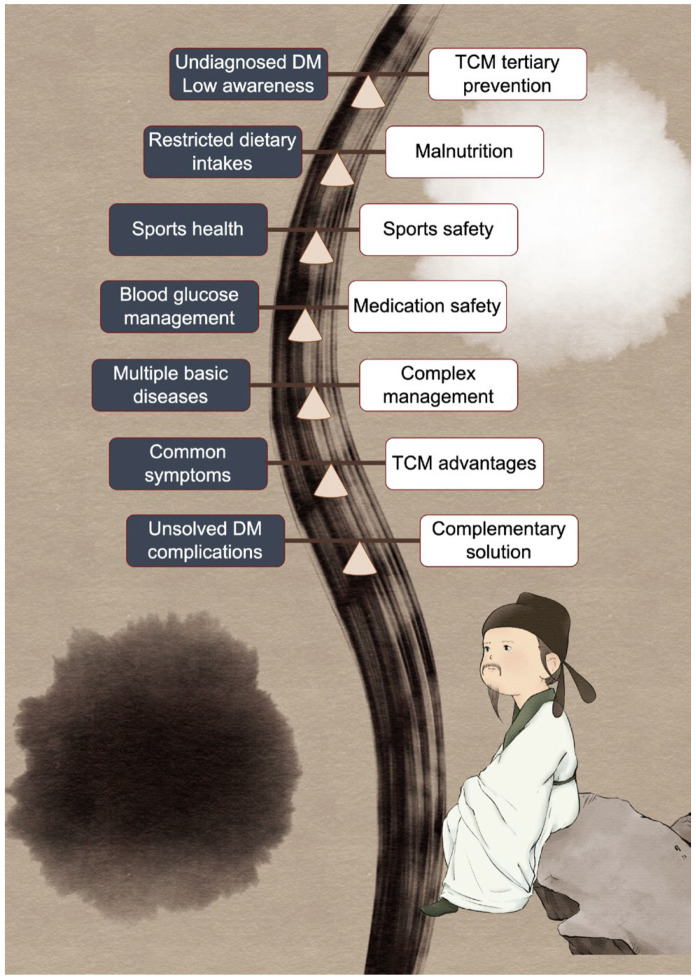
Dilemma in elderly diabetes and the role of TCM and natural products in elderly diabetes. The person used to represent elderly diabetes in this figure is Fu Du, a prominent Chinese poet in the Tang dynasty, who is said to have suffered from diabetes near the end of his life. Yin (black background) and Yang (white background) are two halves, representing the role of traditional Chinese medicine in maintaining body balance. The dilemma mentioned and the related solution are connected by a balance board.

**Figure 2 pharmaceuticals-17-00953-f002:**
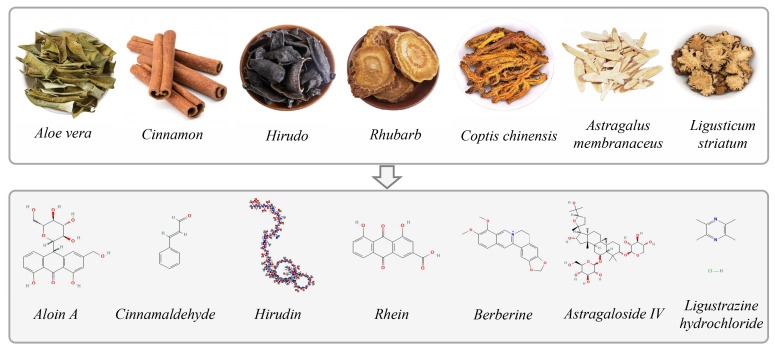
Example of TCM and highly potent chemical compounds used for elderly diabetes and their chemical structure.

**Table 1 pharmaceuticals-17-00953-t001:** Evidence in systemic reviews and meta-analyses on TCM and natural products for the primary prevention of DM.

Ref.	Year	Conditions	Interventions (Dose/Frequency)	Outcomes	No. of Trials	No. of Included Participants
[17]	2009	IGT/impaired fasting blood glucose	Chinese herbal medicines (oral; decoction, 1~3 times daily or one dose every two days, tablet/capsules taken according to the instructions)	More likely to return to normal FBG levels; less likely to progress to diabetes	16	1391
[18]	2016	Prediabetes and T2DM	Aloe vera (oral; Aloe formula 600~1000 mg daily, or raw Aloe leaves 30 g daily)	Improved FPG in prediabetes; improved glycemic control in T2DM, with a marginal improvement in FPG and a significant improvement in HbA1c	8	470
[19]	2016	Diabetes and prediabetes	Fenugreek (seed or powder 5~100 g, capsule 1~5 g, exacts 1 g, total saponins 6.3 g)	Decrease in FBG, 2h BG, HbA1c, and TC	12	1173
[20]	2017	Non-diabetes	TCPM (oral; pill/tablet/capsule/granules taken according to the instructions)	Reduction in FBG, 2h PG, and BMI	26	4169
[21]	2017	IGT	Tianqi capsule (oral; five capsules, tid)	Less likely to progress to T2DM, significantly decreased FBG and 2hBG	6	1027
[22]	2017	Non-diabetes	Different intervention strategies including TCPM (oral; lifestyle intervention, Western medicine, or pill/tablet/capsule/granules taken according to the instructions)	Reduced the rate of incident diabetes	26	10,557
[23]	2018	IGT	TCPM (oral; pill/tablet/capsule/granules taken according to the instructions)	Reduced the incidence of diabetes, normalized the blood glucose, and decreased 2h PG, BMI, fasting insulin, and 2h insulin	18	3172
[24]	2019	Diabetes and prediabetes	Cinnamon (oral; capsules 1~12 g daily, extract 1~14.4 g daily)	Reduced FBG and HOMA-IR	16	1098
[25]	2022	Prediabetes	TCPM (oral; tablet/capsule/granules taken according to the instructions)	Shenqi capsule/granules and Jinqi capsule plus lifestyle modification reduced the incidence rate of DM. Jinlida granules and Tangmaikang granules reduced HbA1c; TCPM, except the Tianqi capsule, reduced FBG; Shenqi capsule/granules reduced PBG	50	4594

Abbreviations: IGT = impaired glucose tolerance; T2DM = type 2 diabetes mellitus; TC = total cholesterol; BMI = body mass index; FBG = fasting blood glucose; HOMA-IR = homeostatic model assessment for insulin resistance; TCPM = traditional Chinese patent medicine; HbA1c = glycated hemoglobin.

**Table 2 pharmaceuticals-17-00953-t002:** Evidence in systemic reviews and meta-analyses on TCM and natural products for diabetic cognitive decline.

Ref.	Year	Conditions	Interventions (Dose/Frequency)	Outcomes	No. of Trials	No. of Included Participants
[37]	2022	T2DM combined with mild cognitive impairment	Adjuvant Chinese medicine (oral; no dose details)	Improved MMSE and MoCA; lower FBG, 2h PG, and HbA1c; improved TC, TG, LDL-C, and HDL-C; reduced TCMSS and incidence of adverse events	16	1299
[38]	2018	Diabetic cognitive impairment	Chinese medicinal herbs (oral; no dose details)	Relieved the symptoms for diabetic cognitive impairment; improved MoCA, MMSE, and TCMSS	9	576
[39]	2023	Diabetes-associated cognitive decline	TCM and Western medicine (oral; no dose details)	Reduced FBG, HbA1c, TNF-α levels, and TCMSS; improved MoCA and MMSE score	20	1570
[40]	2024	Elderly diabetes with cognitive impairment	TCM prescription, TCPM, and TCM extracts (oral; decoction, twice/day, pill/capsule/granules taken according to the instructions)	Improved cognitive function and blood viscosity; improved MMSE and MoCA, CDR and ADL scores	9	848

Abbreviations: TG = triglycerides; LDL-C = low-density lipoprotein; HDL-C = high-density lipoprotein; TCMSS = TCM symptom score; MoCA = Montreal Cognitive Assessment; MMSE = Mini-Mental State Examination; TCPM = traditional Chinese patent medicine.

**Table 3 pharmaceuticals-17-00953-t003:** Evidence in systemic reviews and meta-analyses on TCM and natural products for diabetic sarcopenia, frailty, and obesity in the elderly.

Ref.	Year	Conditions	Interventions (Dose/Frequency)	Outcomes	No. of Trials	No. of Included Participants
[64]	2012	Obesity	Chinese medicine (e.g., white bean extract, Asparagus, Astragali Radix, Coptidis Rhizoma, Rhei Rhizoma, Puerariae Radix, taken according to the guidance) and acupuncture (1~3 times/day, or once/2~3 days, or 1~5 times/week)	Reduced body weight and BMI, with fewer reports of adverse effects and weight regain relapses	96	4861
[65]	2012	Obesity	Mixed oriental herbal medicines (oral; no dose details)	Reduced body weight; improved concomitant conditions including impaired glucose tolerance, hypertension, and inflammation	12	1917
[66]	2013	Obesity	Anti-obesity medicinal plants (e.g., Nigella sativa, Opuntia ficus indica, taken according to the guidance)	Reduced body weight	33	>2344
[67]	2015	Obesity	Acupoint catgut embedding (once/1~15 days, or 1~5 times/week)	Reduced body weight and BMI	43	3520
[68]	2017	Obesity	Acupuncture and lifestyle modification (10~40 min every time, 1~7 days/week)	Reduced body weight and BMI	23	1808
[69]	2018	Obesity	Green tea (oral, 102.5~856.8 mg/d)	Decreased plasma TC and LDL-c levels	21	1704
[70]	2019	Overweight/obesity	Acupuncture (1~7 times/week)	Reduced body weight, BMI, and waist circumference	12	1151
[71]	2019	Overweight or obesity	Quercetin (oral, 100~730 mg/day)	Reduced LDL-c level at doses of ≥250 mg/day and total dose ≥14,000 mg	9	939
[72]	2019	Obesity	Berberine (oral, 0.6~1.7 mg/day)	Reduced TC, TG, and LDL-c levels and elevated HDL-c level	11	1386
[73]	2020	Obesity	Acupuncture (no frequency details)	Reduced body weight, BMI, and body fat mass percentage	8	403
[74]	2020	Obesity	Green tea supplementation (oral, 99~20,000 mg/day)	Reduced body weight, BMI, and waist circumference	26	1344
[75]	2020	Obesity	Acupuncture and related techniques (no frequency details)	Reduced body weight, BMI, and waist circumference; reduced TC and LDL-c levels	33	2503
[76]	2021	Frailty	Sitting Tai Chi (no frequency details)	Improved depressive symptoms, heart rate, and social domain of quality of life of individuals with impaired physical mobility	11	1446
[77]	2022	Obesity	Traditional Kampo medicine bofutsushosan (oral, 2.8 mg/7.5 mg/200 mL, daily)	Reduced BMI	7	679
[78]	2022	Obesity	Anti-obesity herbal medicine (oral; no dose details)	Reduced body weight and BMI	16	1052
[79]	2022	Obesity	Acupuncture and moxibustion therapy (oral; no dose details)	Reduced BMI, body weight, waist circumference, and TG	14	1116
[80]	2022	Obesity	Acupuncture and acupoint catgut embedding (1~7 times/week, or once every 10~14 days)	Reduced body weight, waist circumference, and hip circumference	13	1069
[81]	2022	Obesity	Acupoint catgut embedding, acupuncture (once every 1~30 days)	Reduced body weight and waist circumference	33	2685
[82]	2022	Obesity	TCM (oral, taken according to the guidance)	Reduced body weight, waist circumference, and hip circumference; reduced FBG, TC, and LDL-c levels, and elevated HDL-c level	25	1947
[83]	2022	Obesity	Chinese herbal medicine (oral; no dose details)	Reduced TG and increased HDL-c level	15	1533
[62]	2022	Obesity	Acupuncture (no frequency details)	Improved clinical symptoms; reduced body mass index, FBG, HbA1c, TG, waist circumference, and body fat rate	13	993
[84]	2022	Sarcopenia and frailty	Tai Chi (30~90 min every time, 2~7 times/week)	Improved 30CST, TUGT, number of falls and FOF, SST, balance, DBP, MMSE, depression, and QoL	11	1676
[61]	2022	Sarcopenia	TCM (oral; no dose details)	Improved muscle strength (grip strength, chair stand test) and physical function (6 m walking speed, timed up and go test, sit and reach)	21	1330
[85]	2022	Sarcopenia	TCM (no frequency details)	Improved chair stand test, squatting-to-standing test, 6 m gait speed, timed up and go test, peak torque of the extensors, total work of the extensors, peak torque of the flexors, total work of the flexors, the average power of the flexors, and balance function	13	718
[86]	2023	Obesity	Phytosterol and phytostanol supplementation (oral; 24 mg~3 g/day)	Decreased TC and LDL-c levels	6	337
[87]	2023	Overweight/obesity	Herbal medicines (e.g., Moringa oleifera, oral, taken according to the guidance)	Reduced body weight; change in gut microbiota	7	331
[88]	2023	Obesity	Acupuncture combined with acupoint catgut embedding (30~35 min every time, 1~6 times/week, or once/14 days)	Reduced body weight; change in gut microbiota	20	1616
[89]	2023	Obesity	Cupping therapy (1~5 times/week, or once/10 days)	Reduced body weight, BMI, waist circumference, and hip circumference	21	1563
[90]	2023	Obesity	TCM exercises (e.g., Tai Chi, Qigong, 90~270 min/week)	Reduced body weight, BMI, waist circumference, and fat percentage; improved LDL-c, TG, TC, and HDL-c level	9	1297
[91]	2023	Obesity	TCM exercises (e.g., Tai Chi, Baduanjin, 45~420 min/week)	Reduced body weight, BMI, body fat mass, waist circumference, hip circumference, and waist-to-hip ratio	10	701
[92]	2024	Sarcopenia	Chinese herbal medicine (no frequency details)	Improved total efficiency in sarcopenia; enhanced muscle mass and muscle strength measured by grip strength, 60°/s knee extension peak TQ, muscle function measured by 6 m walking speed, short physical performance battery of 1.50%, EuroQoL 5-dimension	17	1440

30CST = 30 s chair stand test; SST = sit-to-stand test; TUGT = timed up and go test; FOF = fear of falling; DBP = diastolic blood pressure; MMSE = Mini-Mental State Examination; QoL = quality of life.

**Table 4 pharmaceuticals-17-00953-t004:** Evidence included in systemic reviews and meta-analyses on TCM exercise for diabetes and related complications in the elderly.

Ref.	Year	Condition	Interventions (Dose/Frequency)	Outcomes	No. of Trials	No. of Included Participants
[95]	2017	T2DM and DN	Fall prevention programs (including Tai Chi, 1 h twice/week)	Improved balance and walking, without serious AEs	9	423
[96]	2017	T2DM	Baduanjin exercise plus conventional therapy (30~60 min daily, 2~5 times weekly)	Lowered the level of HbA1c, FBG, postprandial plasma glucose, TC, TG, and LDL-C and improved HDL-C, with no mentioned AEs	13	782
[97]	2018	T2DM	Tai Chi (30~60 min daily, 2~7 times weekly)	Lowered FBG and HbA1c	14	798
[98]	2018	T2DM	TCM exercises (Tai Chi, Qigong, and Ba Duan Jin, almost 1 h daily, 2~7 times weekly)	Significantly lowered FBG and HbA1c	39	2917
[10]	2018	T2DM	TCM based lifestyle interventions (Tai Chi and Ba Duan Jin, 20~120 min daily, 1~7 times weekly)	Potentially effective options to improve biomedical and psychosocial well-being, lower FBG, HbA1c, and BMI	24	1697
[99]	2018	DN	Diverse physical rehabilitative interventions (including Tai Chi, 45~60 min daily, 2~7 times weekly)	Improved SF-36 measured QoL (physical function, role physical function, body pain, general health, vitality, social function, role emotional function; mental health, and BMI)	18	1418
[100]	2019	DN	Exercise therapy (including Tai Chi) combined with psychological therapy (the effect of exercise therapy combined with psychological therapy on physical activity and quality of life in patients with painful diabetic neuropathy) (45~60 min daily, 2~7 times weekly)	Improved glucose control, balance, neuropathic symptoms, and some dimensions of QoL	3	>227
[101]	2019	T2DM	Tai Chi (45~60 min daily, 2~7 times weekly)	Lowered FBG, HbA1c, IR, BMI, total cholesterol, blood pressure, and improved QoL-related outcomes (physical function, bodily pain, and social function)	23	1235
[102]	2019	T2DM	Tai Chi (no frequency details)	Lowered FBG and HbA1c	17	754
[103]	2021	T2DM	Tai Chi (no frequency details)	Lowered FBG, TC, HbA1c, and HDL-c	23	1545
[104]	2021	DN	Exercise (including Tai Chi, 20~60 min daily, 3~12 times weekly)	Improved balance and posture	16	883
[105]	2022	T2DM	Baduanjin exercises (20~60 min daily, 5~7 times weekly)	Positive effects on psychological well-being, depression, anxiety, and mental health; improved FBG, HbA1c, and 2h PBG	27	2048
[106]	2022	T2DM	TCM, including Qigong and Tai Chi exercises (5~7 times weekly)	Lowered FBG	17	1052
[107]	2022	T2DM	Tai Chi (no frequency details)	Lowered FBG and improved QoL	7	323
[108]	2022	T2DM	Tai Chi (12~60 min daily, 1~7 times weekly)	Improved FBG, HbA1c, TG, and HDL-C	19	1220
[109]	2023	T2DM	TCM exercise therapy (Taijiquan, Baduanjin, Yijinjing, and Wu Qin Xi, 12~60 min daily, 1~7 times weekly)	Improved blood glucose levels, blood lipid levels, and insulin-related indicators	33	2609
[110]	2023	T2DM	Baduanjin exercises (12~60 min daily, 1~7 times weekly)	Alleviated HbA1c, depression, and anxiety	11	755
[111]	2023	T2DM	Tai Chi (30~180 daily, 1~7 times weekly)	Improved FBG, HbA1c, TG, and TC	16	1212
[112]	2023	T2DM	Mind–body exercises (including Tai Chi, no frequency details)	Improved blood glucose and lipid levels	52	4024
[113]	2024	T2DM	Therapeutic interventions to improve static balance (including Tai Chi, 2~113 min daily, 1~7 times weekly)	Increased time in the one-leg stance, Romberg test, and tandem position, increased the Berg Balance Scale score and balance index, and reduced the variables of postural sway	31	1821
[114]	2024	T2DM	Tai Chi (no frequency details)	Improved HbA1c, FBG, BMI, and overall QoL	17	12,720

Abbreviations: AEs = adverse effects; BMI = body mass index; FBG = fasting blood glucose; HbA1c = hemoglobin A1C; HDL-C = high-density lipoprotein cholesterol; IR = insulin resistance; QoL = quality of life; TC = total cholesterol; TCM = traditional Chinese medicine; TG = total triglyceride; 2h PBG = 2 h post-load glucose.

**Table 5 pharmaceuticals-17-00953-t005:** Evidence included in systemic reviews and meta-analyses on TCM and natural products for blood glucose control in elderly diabetes.

Ref.	Year	Condition	Interventions (Dose/Frequency)	Outcomes	No. of Trials	No. of Included Participants
[40]	2024	Elderly diabetes	TCM (TCM prescription, TCM patent medicines, TCM extracts) (oral; decoction, 1~3 times daily, pill/capsule/granules taken according to the instructions, Corn silk aqueous extract 2.4 g, twice/day)	Improved blood lipid metabolism, islet function, and insulin resistance	28	2737

**Table 6 pharmaceuticals-17-00953-t006:** Evidence included in systemic reviews and meta-analyses on TCM and natural products for elderly diabetes combined with cardiovascular disease.

Ref.	Year	Condition	Interventions (Dose/Frequency)	Outcomes	No. of Trials	No. of Included Participants
[129]	2021	DM and CHD	Danhong injection (intravenous; 20~40 mL/day)	Improved the comprehensive clinical effectiveness rate, ECG efficiency, efficiency of angina pectoris, cholesterol level, LDL-c, coronary angina attack frequency, and duration of angina pectoris	14	1454
[130]	2022	DM	Camellia tea and herbal tea (oral; different types of tea 0.25~7.5 g daily, extract 150~750 mg daily)	Regulatory effects on adipose, glycemic control, lipid profiles, blood pressure, and cardiometabolic risk	19	832
[131]	2022	DM and CHD	TCM (oral; no dose details)	Improved the cardiac function, hemorheology, blood glucose, blood lipids, and inflammation, thus reducing the frequency of angina, the incidence of cardiovascular events, and all-cause mortality	20	3565
[132]	2023	DM and CHD	TCM injection (intravenous; used according to the instructions)	Improved the total effectiveness rate, electrocardiogram effectiveness rate, and effective rate of angina	53	4619
[133]	2023	DM	Fufang Danshen Dripping Pill (oral; 270 mg daily or taken according to the instructions)	Reduced low shear rate, high shear rate, plasma viscosity, and homocysteine, and increased plasma adiponectin	18	1532
[134]	2023	DM and angina pectoris	Oral Chinese patent medicines (oral; pill/capsule, taken according to the instructions)	Reduced the incidence of cardiovascular AEs, the frequency and duration of angina pectoris, FBG, 2h PBG, and HbA1c, blood lipid TC and LDL-C, and TG	45	4727

Abbreviations: CHD = coronary heart disease; DM = diabetes mellitus; ECG = electrocardiograph; LDL-c = low-density lipoprotein cholesterol.

**Table 7 pharmaceuticals-17-00953-t007:** Evidence in systemic reviews and meta-analyses on TCM and natural products for the management of common diabetic complications in elderly diabetes.

Ref.	Year	Condition	Interventions (Dose/Frequency)	Outcomes	No. of Trials	No. of Included Participants
[149]	2011	DKD	Astragalus membranaceus (intravenous, 20~60 mL/day)	Reduced BUN, SCr, CCr, and UTP; improved systemic state	25	1804
[150]	2012		Ligustrazine (intravenous, 40–400 mg/day	Reduced BUN, SCr, 24h UTP, UALB, and UAER	25	1645
[151]	2015		Flos Abelmoschus Manihot (oral, extract, 1.2 g/day, Huangkui capsule 7.5 g/day)	Decreased proteinuria and serum creatinine	7	531
[152]	2015		Flos Abelmoschi Manihot (oral; no dose details)	Improved serum albumin level; reduced BUN, SCr, 24h UTP, and UAER	27	2239
[153]	2019		Tripterygium wilfordii (oral; 10~20 mg, three times/day, or 0.5~2 mg/kg/d)	Reduced 24h UTP; increased serum albumin	22	1414
[154]	2019		Safflower yellow (intravenous, 2~150 mL, or 100~200 mg/day)	Lessened UAER, heightened the proportion of blood sugar, and improved other detection indicators related to DKD	18	1289
[155]	2019		Astragalus membranaceus (intravenous, 20~60 mL, daily; decoction, 30~60 g/day; granules, 8 g/day; tablet, eight pieces/day)	Reduced more UTP and SCr levels	66	4785
[156]	2019		Chinese herbal medicine (oral; tablet/capsules taken according to the instructions)	Reduced UTP	20	2719
[157]	2019		Buyang Huanwu decoction (oral; decoction, no dose details)	Reduced UAER and TC	15	1402
[158]	2020		Bailing capsules (oral; 150 mg~6 g/day, or 6~15 pellets/day)	Reduce 24h UTP, UAER, SCr, and BUN levels	24	1941
[159]	2020		Panax notoginseng preparations (oral; no dose details)	Reduced UTP, SCr, TC, TG, and LDL-c	24	1918
[160]	2021		Tripterygium glycosides (oral; no dose details)	Reduced 24h TUP, elevated serum albumin	26	1824
[161]	2021		Tripterygium glycosides (oral; 10~60 mg/d, or 1~2 mg/(kg·d))	Decreased 24h UTP and SCr levels	31	2764
[162]	2021		*Salvia miltiorrhiza* Bunge and ligustrazine injection (intravenous, 5~20 mL, daily)	Reduced UAER, SCr, and β2-MG; reduced IL-6, IL-18, and TNF-α	21	1939
[163]	2021		Yiqi Huoxue prescription (oral; no dose details)	Lowered UAER, SCr, BUN, FBG, HbA1c, TG, and TC	13	1332
[164]	2021		TCM (oral; no dose details)	Decreased SCr, BUN, UAER, 24h UTP, and TNF-α, and improved high-sensitivity CRP	20	3566
[165]	2022		Tripterygium glycosides (oral; 30~90 mg/d, or 0.5~2 mg/(kg·d))	Reduced UAER, 24h UTP, and SCr; increased albumin level	33	2034
[166]	2022		TCM (oral; no dose details)	Reduced 24h UTP, SCr, and BUN	56	5464
[167]	2022		Chinese patent medicines (oral; tablet/capsules taken according to the instructions)	Reduced UAER, 24h UTP, SCr, BUN, TG, TC, and CRP	62	5362
[168]	2023		Dihuang pill prescriptions (oral; no dose details)	Reduced SCr, 24h UTP, FBG, and UAER	41	3562
[169]	2023		Chinese patent medicine (oral; no dose details)	Reduced SCr, BUN, and UAER	53	4891
[170]	2023		Astragalus membranaceus and Rhizoma Dioscoreae (oral; no dose details)	Reduced 24h UP	7	556
[171]	2023		Qi-supplementing and Yin-nourishing Chinese patent medicines (oral; no dose details)	Reduced UAER, SCr, 24h UTP, blood sugar, and blood lipids	72	6344
[172]	2023		Ophiocordyceps sinensis (oral; tablet/capsules taken according to the instructions)	Reduced SCr, BUN, β2-MG, CysC, 24h UTP, UALB, UAER, and ACR	38	3167
[173]	2023		Danggui Buxue decoction (oral; Radix Astragali 30~45 g, Angelicae Sinensis 6~9 g)	Reduced 24h UTP, SCr, BUN, blood glucose and lipid levels; improved clinical outcomes; modulated inflammatory factor levels	6	472
[174]	2024		Acupuncture (20~35 min, once a day or once every 2 days)	Increased the clinical effectiveness rate; reduced UTP, UALB, β2-MG, SCr, HbA1c, FBG, 2h PBG, TC, TG, and HDL-c	9	659
[175]	2024		Salvia miltiorrhiza and ligustrazine injection (intravenous, 10~20 mL or 40 mg daily)	Reduced SCr, BUN, β2-MG, UTP, UAER, ACR, 24h UTP, and UALB; improved blood glucose, blood pressure, lipids, and inflammatory responses	30	3214
[176]	2018	DR	Single herbal medicine (e.g., Ruscus extract tablet, Sanqi Tongshu capsule, tetramethylpyrazine injection, Xueshuantong injection, puerarin injection, and Xuesaitong injection; oral or intravenous, used according to the instructions)	Increased the chances of visual improvement; reduced blood vessel bleeding and blood sugar level	10	754
[177]	2020		Acupuncture (no frequency details)	Improved visual acuity	6	864
[178]	2020		TCM (e.g., Tongmai Zengshi capsule, Tangmuning decoction, Fufang Xue Shuan Tong capsule) (oral or intravenous; decoction, twice daily; tablet/capsule/pill/granules/injection, used according to the instructions; Panax Notoginseng powder 2 g, three times/day)	Improved visual acuity, micro-aneurysms, and HbA1c	33	3373
[179]	2021		Compound Danshen Dripping Pills (oral; 10 pills/time, three times/day)	Improved visual field gray value, hemangioma volume, hemorrhagic plaque area, and visual acuity	8	524
[180]	2022		Chinese herbal compounds (e.g., Bushen Huoxue Mingmu Tang, Ziyin Mingmu decoction, Yiqi Yangyin experience formula) (oral; no dose details)	Improved clinical efficacy and best corrected visual acuity, reducing the number of microangiomas, the microangioma volume, hemorrhagic spots, hemorrhagic area, hard exudates, cotton lint spots, central macular thickness, Chinese medicine evidence score, FBG, 2h PBG, HbA1c, TC, and less adverse events occurred	27	2144
[181]	2022		TCM injections, including Astragalus, danhong, ginkgo biloba extract powder, ginkgo leaf extract and dipyridamole, ligustrazine, Mailuoning, puerarin, safflower, Shuxuetong, safflower yellow sodium chloride, and Xueshuantong (oral; no dose details)	Improved clinical effectiveness and vision	45	4134
[182]	2022		Chinese patent medicines, including compound Xueshuantong, compound Danshen Dripping Pill, and Shuangdan Mingmu capsule (oral/intravenous; capsule/pill/injection, used according to the instructions)	Improved retinal microaneurysm, hemorrhage, macular thickness, visual acuity, FBG, and HbA1c	19	1568
[183]	2023		Combined herbal adjuvant therapy involved 69 kinds of traditional Chinese medicine, such as Panax notoginseng, Rehmannia rehmannii, Astragalus membranaceus, and Poria cocos (oral; no dose details)	Improved clinical effectiveness rate, visual acuity, fundus efficacy, neovascularization regression rate, macular foveal thickness, absorption of vitreous hemorrhage, FBG, and 2h PBG	18	1392
[184]	2023		Oral Chinese patent medicines, including Xueshuantong capsule, Danshen Dripping Pill, Shuangdan Mingmu capsule, Qiming granules, Hexuemingmu tablet, Mingmu Dihuang pill (oral; no dose details)	Improved visual acuity and visual field gray value; reduced microaneurysm volume, hemorrhage area, and macular thickness	42	4858
[185]	2011	DPN	Chinese herbal medicine, including four single herbs, eight traditional Chinese patent medicines, and 18 self-concocted Chinese herbal compound prescriptions (oral; 10 pills/time, three times/day)	Improved global symptoms (including improvement in numbness or pain) and changes in nerve conduction velocity	39	2890
[186]	2012		Chinese herbal medicine involving Astragalus Radix, Angelicae Sinensis Radix, Pheretima, Chuanxiong Rhizoma, Codonopsis Radix, Carthami Flos, Hirudo, Rehmanniae Radix, Spatholobi Caulis, Paeoniae Radix Rubra, Cyathulae Radix, Asari Radix ET Rhizoma, and Scolopendra (oral; no dose details)	Effective in treating DPN	18	1575
[187]	2013		Chinese herbal medicine, including single herbs, Chinese traditional patent medicines, and self-composed Chinese herbal compound prescriptions (e.g., modified Huang Qi Gui Zhi Wu Wu Tang, Tongxinluo capsule, modified Buyang Huanwu Tang, formulations of Xuesaitong) (oral/intravenous; capsule/pill/injection, used according to the instructions)	Improved global symptoms (including improvement in numbness or pain) and changes in nerve conduction velocity	49	3639
[188]	2013		Chinese herbal medicine (e.g., Guizhi Shaoyao Zhimu Tang, Xiaoke Tongluo capsule, Yiqi Wenyang Huoxue Tang) (oral; decoction, three times/day; capsule, taken according to the instructions)	Improved clinical symptoms	10	653
[189]	2014		Puerarin injection (intravenous; 300~500 mg, or 30 mL daily)	Improved the total effectiveness rate, correct nerve conduction velocity that was decreased by diabetes, and improved the hemorheology index	22	1664
[190]	2017		Chinese medicine based on Yang-warming method (oral; no dose details)	Improved the nerve conduction velocity	25	1203
[191]	2020		Traditional Chinese medicine foot bath, combined with acupoint massage (1~2 times/day)	Improved the total effectiveness rate, SNCV, MNCV, and neuropathic syndrome score	31	3284
[192]	2020		Manual acupuncture (no frequency details)	Improved clinical efficacy and nerve conduction velocity, and median nerve SCV, and decreased the scores in the Toronto clinical scoring system	11	1200
[193]	2020		Moxibustion (no frequency details)	Increased median MNCV, peroneal MNCV, SNCV, peroneal SNCV, and total effectiveness rate	11	927
[194]	2022		Herbal medicines (oral; no dose details)	Increased SNCV, MNCV of the median and common peroneal nerves, and improved TER	72	6260
[195]	2022		Shuxuetong injection (intravenous; no dose details)	Improved the clinical symptoms, SCV, MCV, and average blood glucose	6	507
[196]	2023		Buyang Huanwu decoction (oral; twice/day)	Improved the median nerve MNCV and SNCV; decreased plasma viscosity, whole blood high shear rate, and whole blood low shear rate	21	1945
[197]	2023		Tangmaikang granules (oral; no dose details)	Increased clinical effectiveness rate and nerve conduction velocity, as well as improved the symptoms of the peripheral nerve, and the blood glucose level	19	1602
[198]	2024		*Radix Astragali Mongolici*-based TCM (oral; no dose details)	Decreased the Toronto Clinical Scoring System scores; reduced glycaemia, IL-6, and TNF-α levels; increased nerve conduction velocity	48	3759
[40]	2024		TCM (oral; decoction, twice/day daily, berberine 700 mg, three times/day)	Improved FBG, 2h PBG, HbA1c, serum ferritin, superoxide dismutase, Glutathione peroxidase, SNCV, MNCV, TC, TG, LDL-c, and HDL-c	5	446

Abbreviations: ACR = albumin–creatinine ratio; β2-MG = β2-microglobulin; BUN = blood urea nitrogen; CHD = coronary heart disease; CRP = C-reactive protein; CysC = cystatin C; DKD = diabetic kidney disease; DM = diabetes mellitus; DPN = diabetic peripheral neuropathy; DR = diabetic retinopathy; ECG = electrocardiography; FBG = fasting blood glucose; HbA1c = hemoglobin A1c; HDL-c = high-density lipoprotein cholesterol; MNCV = motor nerve conduction velocity; SCr = serum creatinine; SNCV = sensory nerve conduction velocity; TC = total cholesterol; TCM = traditional Chinese medicine; TG = total triglyceride; TNF-α = tumor necrosis factor-alpha; UAER = urinary albumin excretion rate; UALB = urine microalbumin; UTP = urinary total protein; 2h PBG = 2h post-load glucose.

## Data Availability

Data sharing is not applicable to this article as no new data were created or analyzed in this study.

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
