# Peer review of "Dilemmas in Elderly Diabetes and Clinical Practice Involving Traditional Chinese Medicine"

_pharmaceuticals, 2024, doi:10.3390/ph17070953_

Round 1

Reviewer 1 Report

Comments and Suggestions for Authors

The review “Dilemmas in elderly diabetes and clinical practice with traditional Chinese medicine” describes the authors look at the difference in the approach to treat young and elderly patients with diabetes in China by use of traditional medicine. The review seems to be actual, opens up needed discussion of the personalized medicine. The paper could be accepted after minor revision. The authors should name  all “Dilemmas” point by point according to the title of the manuscript. The sections 8 and 9 should be more focused and contain precise conclusions but not general formulations. What could be an influence of TCM on the world approaches to treat diabetes for young and elderly people? Are there any general approaches in China and Western world? What are the most effective TCM that could be recommended to elderly people in Western world? See also for the typos (for ex, Lines 660-662).

Author Response

Comment 1: The review “Dilemmas in elderly diabetes and clinical practice with traditional Chinese medicine” describes the authors look at the difference in the approach to treat young and elderly patients with diabetes in China by use of traditional medicine. The review seems to be actual, opens up needed discussion of the personalized medicine. The paper could be accepted after minor revision.

Response 1: We thank the reviewer for the positive comments on our work.

Comment 2: The authors should name all “Dilemmas” point by point according to the title of the manuscript.

Response 2: As requested, we have named all “Dilemmas” point by point and revised those subtitles in the revised manuscript, as easily found in the document with the changes highlighted (section 1-7).

Comment 3: The sections 8 and 9 should be more focused and contain precise conclusions but not general formulations.

Response 3: We thank the reviewer for the kind suggestion. The sections 8 and 9 have been comprehensively sorted out, with a view to providing precise conclusions to the development of the research on TCM and natural products. We provided suggested research directions (Line 323~369) including: (1) Development of novel elderly diabetes patients-centered TCM and natural products treat-ment. (2) Transition from experience-based to evidence-based approaches in TCM and natural products. (3) Clinical trials. (4) Explore a wider range of study methodologies. (5) Employ novel evaluation standards tailored to the unique features of TCM. (6) Personalized treatment. (7) Multidisciplinary and multilevel research approaches. (8) Overseas development of TCM. (9) Research on the active pharmacodynamic material basis of TCM. (10) More comprehensive and high-quality verification experiments.

Comment 4: What could be an influence of TCM on the world approaches to treat diabetes for young and elderly people?

Response 4: Thank you for the kind suggestion. As requested, we have added more discussion on this issue (section Introduction and section 8-9). As described in our revised manuscript, ethnopharmacology-focused research endeavors not only offer a scientific foundation for determining optimal dosages and potential toxicological impacts within local communities but also hold promise for the development of more efficacious multitarget pharmaceuticals aimed at pre-venting and treating a range of ailments, including diabetes, in elderly people. Learning from the successful research experience of Artemisinin, selection of effective drug targets from TCM and natural products under the guidance of unique TCM theory accelerates the drug discovery process. TCM and natural products also play a role in enhancing the effectiveness of an-tidiabetes medications and in the prevention and management of complications, offering a novel perspective. For instance, Qi-tonifying herbs could promote energy metabolism to control weight and boost immunity against diabetic complications. Existing evidence demonstrates that TCM and natural products improve the prevention of diabetes in elderly individuals, keep nutritional balance and sport health, manage clinical symptoms, control and even reverse diabetes and diabetic complications, provide a complementary and alternative therapy.

Comment 5: Are there any general approaches in China and Western world? What are the most effective TCM that could be recommended to elderly people in Western world?

Response 5: Thank you for the valuable suggestion, and we have added this comment in the section 8 (7). In the process of drug discovery, crucial mechanism of diabetes provides directions for targets selection. For example, the conceptualization of Gluco-Cardio-Renal conditions guide researchers to explore multi-targets antidiabetes medications, while that's exactly what TCM is good at. Acceptable TCM therapies (especially acupuncture, well-defined oral or topical TCM and natural products, TCM exercise) help to build trust in TCM among the general public and healthcare professionals. As for selection of the best treatment, well-designed network meta-analysis may give the final answer, which could be conducted in the future. In this review, we aims to provide evidence-based options to choose, while clinical decision-making should be personal and comprehensive. Various medical disciplines involved treatment plans may enhance the therapeutic effect.  

Comment 6: See also for the typos (for ex, Lines 660-662).

Response We thank the reviewer for catching these mistakes, which have been corrected. (line 376-381)

Reviewer 2 Report

Comments and Suggestions for Authors

The review work entitled “Dilemmas in elderly diabetes and clinical practice with traditional Chinese medicine” by Xue et al. highlights the role of traditional Chinese medicine in diabetes care and future development prospects. The work is well-complied, providing detailed insights on the current clinical practices and the benefits and limitations of TCM. Here are a few suggestions for the improvement of the work for authors before consideration of the work:

 Please define the objective and scope in the introduction for consistent alignment throughout the manuscript.

Provide a balanced discussion of the benefits and limitations of TCM and natural products, addressing potential biases. It would be great if a few graphical figures could be included in the study. Also, try to draw the chemical structures of a few highly potent chemical compounds.

It would be great if data on the dose and mode of action of the intervention were included in the table.

Use clear and concise language throughout the paper, avoiding complex sentences for better readability.

Please strengthen the conclusion by summarizing key points and clearly stating the implications and recommendations for future research.

Comments on the Quality of English Language

Use clear and concise language throughout the paper, avoiding complex sentences for better readability.

Author Response

The review work entitled “Dilemmas in elderly diabetes and clinical practice with traditional Chinese medicine” by Xue et al. highlights the role of traditional Chinese medicine in diabetes care and future development prospects. The work is well-complied, providing detailed insights on the current clinical practices and the benefits and limitations of TCM. Here are a few suggestions for the improvement of the work for authors before consideration of the work:

Comment 1: Please define the objective and scope in the introduction for consistent alignment throughout the manuscript.

Response 1: As requested, we have revised the manuscript to add emphasis on these aspects. (Line 65-71)

Comment 2: Provide a balanced discussion of the benefits and limitations of TCM and natural products, addressing potential biases. It would be great if a few graphical figures could be included in the study. Also, try to draw the chemical structures of a few highly potent chemical compounds.

Response 2: Thank you for the valuable suggestion. We have added the discussion on the limitations of TCM and natural products to address potential biases (Line 323-332). Also, we summarized highly potent chemical compounds used for elderly diabetes and added their chemical structures in Figure 2.

Comment 3: It would be great if data on the dose and mode of action of the intervention were included in the table.

Response 3: We thank the reviewer for this kind comment. We checked all the data included in these tables to supply relavant dose and mode of action of the intervention. 

Though we have added detailed data to these tables as we can, the dose-effect relationship of TCM and natural products for elderly diabetes and medication guides still need more precise and credible conclusions due to the absence of data. Thus, more articles published with complete data would be needed in the future to address these issues.

Comment 4: Use clear and concise language throughout the paper, avoiding complex sentences for better readability.

Response 4: We thank the reviewer for the positive comments about our work. We made every effort to ensure the writing quality of our article. Two co-authors (Chongxiang Xue and Ying Chen) conducted a grammatical check for this manuscript to avoid grammatical errors or typos and a third co-author (Han Wang) checked the use of professional terminology. Also, this revised manuscript had been edited by native speakers to ensure the accuracy of the language. We hope that there will be no barriers to comprehension for readers.

Comment 5: Please strengthen the conclusion by summarizing key points and clearly stating the implications and recommendations for future research.

Response 5: Thank you for the valuable suggestion. The part of the conclusion has been comprehensively sorted out, with a view to providing precise conclusions to the development of the research on TCM and natural products. 

Reviewer 3 Report

Comments and Suggestions for Authors

This manuscript outlines the challenges and dilemmas associated with treating elderly patients with diabetes, addressing issues such as the importance of early diagnosis and prevention, as well as the delicate balance between restricted diet intake and the risk of undernutrition. It highlights the diverse treatment approaches of Traditional Chinese Medicine (TCM) that bolster the effectiveness of diabetes prevention and management in elderly individuals, streamlining medication administration and alleviating common symptoms and complications. Furthermore, it delves into the historical significance of natural products within TCM and their potential for facilitating drug discovery in diabetes management, notwithstanding the limited conclusions reached on this subject. Following the revisions made to this manuscript, notable improvements have been achieved. Overall, this manuscript exhibits commendable organization and clarity of writing.

Author Response

Comment 1: This manuscript outlines the challenges and dilemmas associated with treating elderly patients with diabetes, addressing issues such as the importance of early diagnosis and prevention, as well as the delicate balance between restricted diet intake and the risk of undernutrition. It highlights the diverse treatment approaches of Traditional Chinese Medicine (TCM) that bolster the effectiveness of diabetes prevention and management in elderly individuals, streamlining medication administration and alleviating common symptoms and complications. Furthermore, it delves into the historical significance of natural products within TCM and their potential for facilitating drug discovery in diabetes management, notwithstanding the limited conclusions reached on this subject. Following the revisions made to this manuscript, notable improvements have been achieved. Overall, this manuscript exhibits commendable organization and clarity of writing.

Response 1: We thank the reviewer for the positive comments on our work.

Round 2

Reviewer 2 Report

Comments and Suggestions for Authors

The work is now improved significantly and therefore can be considered for the publication.